# DP-ImgSyn: Dataset Alignment for Obfuscated, Differentially Private Image Synthesis

## Abstract

The availability of abundant data has catalyzed the expansion of deep learning vision algorithms. However, certain vision datasets depict visually sensitive content such as content moderation images. Sharing or releasing these datasets to the community would improve the performance of neural models, but poses moral and ethical questions. Thus, there is a need to share such datasets with privacy guarantees without sharing visually sensitive data. Traditionally, Generative Adversarial Networks (GANs) with Differential Privacy (DP) guarantees are employed to generate and release data. However, GAN-based approaches result in images that are visually similar to private images. In this paper, we propose a non-generative framework, Differentially Private Image Synthesis (DP-ImgSyn), to sanitize and release visually sensitive data with DP guarantees to address these issues. DP-ImgSyn consists of the following steps. First, a teacher model is trained (for classification) using a DP training algorithm. Second, optimization is performed on a public dataset using the teacher model to align it with the private dataset. We show that this alignment improves performance (up to $\approx$ **17%**) and ensures that the generated/aligned images are visually similar to the public images. The optimization uses the teacher network's batch normalization layer statistics (mean, standard deviation) to inject information about the private images into the public images. The synthesized images with their corresponding soft labels obtained by teacher model are released as the sanitized dataset. A student model is trained on the released dataset using KL-divergence loss. The proposed framework circumvents the issues of generative methods and generates images visually similar to the public dataset. Thus, it obfuscates the private dataset using the public dataset. Our experiments on various vision datasets show that when using similar DP training mechanisms, our framework performs better than generative techniques (up to $\approx$ **20%**).

## 1 Introduction

Deep Learning has benefited from large statistical data made available to the broad community. Large datasets in the domain of image classification (Deng et al., 2009), object recognition (Lin et al., 2014), language modeling (Lewis et al., 2004), and recommendation systems (Bennett et al., 2007) have helped these domains make significant advances. However, there are cases in which vision datasets depict visually disturbing and sensitive content. For example, content moderation (Steiger et al., 2021) data or child sexual abuse material (CSAM Apple, 2021; Cobbe, 2021). In such cases sharing or releasing these datasets for the community to improve the performance of models poses moral and ethical questions. Thus, there is a need to release sensitive datasets while achieving visual dissimilarity between the private and the released data.

To alleviate this problem, a popular solution is to release images generated by Generative Adversarial Networks (GANs Goodfellow et al., 2014) with Differential Privacy (DP) guarantees (DP-GANs Xie et al., 2018; Xu et al., 2019; Zhang et al., 2018; Xie et al., 2018; Cao et al., 2021). However, GAN-based methods for generating synthetic images result in images that are visually similar to private images. This is a severe limitation when private images depict sensitive content. In addition to that, GAN training faces multiple challenges (Arjovsky & Bottou, 2017) such as vanishing gradients (Arjovsky & Bottou, 2017), mode collapse

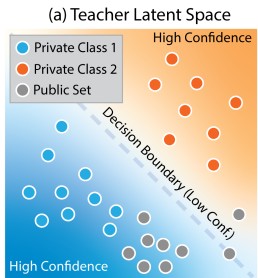 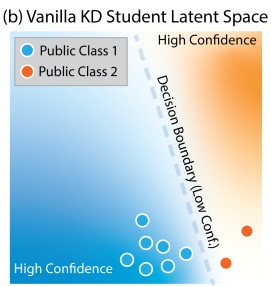 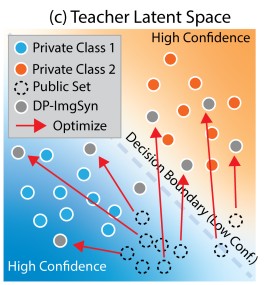 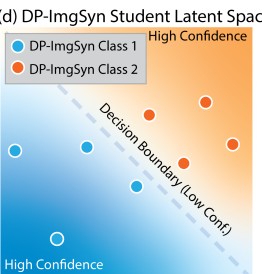

Figure 1: (a) illustrates a cartoon version of the decision boundary of the teacher model and how a public dataset (gray) samples the latent space. (b) shows the transfer of the decision boundary of the teacher model in (a) to the student model using the public images to sample the latent space. We see that the student in (b) learns a decision boundary different from the teacher leading to poor knowledge transfer. (c) shows the effect of optimization/perturbation of the public images with DP-ImgSyn. With DP-ImgSyn the latent space is sampled more effectively. Finally, (d) illustrates the transfer of the decision boundary of the teacher model to the student model, using synthetic images generated by DP-ImgSyn to sample the latent space.

(Srivastava et al., 2017; Brock et al., 2019; Arjovsky et al., 2017; Salimans et al., 2018; Miyato et al., 2018), training instability (Brock et al., 2019) and convergence failure (Salimans et al., 2016).

In this paper, we propose a non-generative (or discriminative) approach for sanitizing and releasing datasets while maintaining visual privacy to address the above-mentioned challenges. Specifically, we consider the following task of *generating a synthetic dataset with the following three requirements (1) The synthetic dataset must be $(\epsilon, \delta)$ differentially private. (2) The synthetic dataset must be visually dissimilar to the public dataset, i.e. generating visually non-sensitive when the private dataset is such. (3) The synthetic dataset has a similar utility as the private dataset in the downstream task of classification.*

To meet these requirements we propose our framework, Differentially Private Image Synthesis (DP-ImgSyn), trains a teacher model on the private (sensitive) dataset using a DP training algorithm, such as DP-SGD (Abadi et al., 2016). Next, a public dataset is selected for distillation (Hinton et al., 2015) in which we distill the sensitive dataset. We observe that misalignment between the public-private datasets can result in significant performance degradation. To address dataset misalignment, we propose an alignment technique on the public dataset to improve the performance of our framework on misaligned public-private dataset pairs. Finally, we obtain soft labels using the teacher model and the aligned public dataset. The soft label is the teacher model output (logits) after applying the softmax function. The aligned public dataset and the corresponding DP-teacher model generated soft labels are released as the sanitized dataset.

Our proposal is based on two key ideas. Firstly, the aim is to learn/transfer the decision boundaries between the teacher and the student model. This is done by sampling the latent space using the public images and having the student model match the decision boundary values using soft labels. This implies that latent space sampling needs to be effective. The challenge with this is that public images at the neural network input might not effectively sample the latent space. Thus, the second key idea is our proposed optimization/modification whose goal is to perturb the public images such that the latent space can be effectively sampled. This leads to better knowledge transfer between the teacher and the student model. Figure 1 (a) illustrates a cartoon version of the decision boundary of the teacher model and how a public dataset (gray) samples the latent space. Figure 1 (b) shows the transfer of the decision boundary of the teacher model in 1 (a) to the student model using the public images to sample the latent space. We see that the student in 1 (b) learns a decision boundary different from the teacher leading to poor knowledge transfer. Figure 1 (c) shows the effect of optimization/perturbation of the public images with DP-ImgSyn. With DP-ImgSyn the latent space is sampled more effectively. Finally, Figure 1 (d) illustrates the transfer of the decision boundary of the teacher model to the student model, using synthetic images generated by DP-ImgSyn to sample the latent space. For actual visualizations on deep nets please see Appendix A.3.

Our framework avoids the challenges that generative techniques such as GANs (Goodfellow et al., 2014) face by leveraging distillation and reducing public-private distribution misalignment. This alignment improves

Private Dataset: CelebA Hair 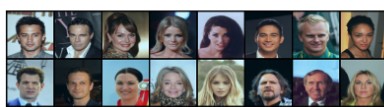    DP - ImgSyn 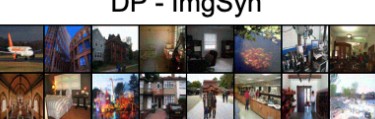    Public Dataset: Places365 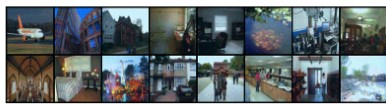

Figure 2: DP-ImgSyn synthesized images are visually dissimilar to private images. Visualization of CelebA-Hair (private dataset) and the DP-ImgSyn generated sanitized dataset (middle). DP-ImgSyn is initialized with Places365 as public dataset (right). A model trained on the synthetic CelebA-Hair (DP-ImgSyn generated) dataset achieves ≈ 99.5% the performance of a model trained with DP on the CelebA-Hair dataset. See extended version in Figure 6.

performance up to ≈ **17%** on highly misaligned public-private dataset pairs. Further, DP-ImgSyn synthesizes images that are visually dissimilar to private images. This can be observed in Figure 2, which visualizes the private dataset (CelebA-Hair left), the corresponding sanitized dataset (synthetic CelebA-Hair middle) generated by DP-ImgSyn, and the public dataset used for initialization (right). A model trained on the synthetic CelebA-Hair dataset achieves ≈ 99.5% the performance of a model trained with DP on the CelebA-Hair dataset. Since all the mechanisms used are DP guaranted, the released images are also DP guaranteed to the same bounds. We experimentally evaluate the synthetic images in terms of classification accuracy on MNIST, FashionMNIST, CelebA-Hair, CelebA-Gender, CIFAR10, and ImageNette datasets. Our method achieves significantly better accuracy (up to ≈ **20%**) than state-of-the-art generative methods using a similar DP training algorithm. We wish to emphasize that the proposed technique is not a new DP mechanism but is rather a new approach to visual privacy for sensitive datasets which leverages dataset alignment to obfuscate sensitive data in public datasets.

Our contributions are summarized as follows:

- We propose DP-ImgSyn, a non-generative image synthesis framework that generates a DP-guaranteed dataset for public release. The synthetic images satisfy two properties: 1) visual dissimilarity to private images and, 2) are DP-guaranteed.

- The DP-ImgSyn framework leverages the teacher model's batch normalization layer statistics to address the distribution misalignment between private and public datasets.

- We show the effectiveness of DP-ImgSyn in image classification tasks on various vision datasets. We also show that DP-ImgSyn performs better than state-of-the-art generative methods when using a similar DP training algorithm.

## 2 Background and Related Work

This section briefly introduces the definition of Differential Privacy (DP), followed by an overview of research initiatives that address dataset sanitization for public release. Differential Privacy (DP) was introduced by Dwork et al. (2006). A randomized algorithm $\mathcal{A}$ is said to be $(\epsilon, \delta)$ differentially private, if for all $\mathcal{S} \subseteq \text{Range}(\mathcal{A})$ and for all datasets $x, y \in \text{Domain}(\mathcal{A})$ such that $||x - y||_1 \leq 1$:

$$\Pr\left[\mathcal{A}(x) \in \mathcal{S}\right] \leq e^\epsilon \cdot \Pr\left[\mathcal{A}(y) \in \mathcal{S}\right] + \delta$$

In the case of deep learning, $\mathcal{A}$ is the training algorithm, and $\mathcal{S}$ is the subset of all possible model parameters that can be output from the training process.

Regarding sanitizing datasets for public release, generative techniques for DP data release are widely used. They leverage GANs or other generative models by sharing DP-trained models, embeddings, or generating images for releasing datasets while maintaining privacy. Multiple research articles (Xie et al., 2018; Xu et al., 2019; Zhang et al., 2018) have proposed training GANs with DP for image synthesis using DP-SGD (Abadi et al., 2016) under various contexts and domains. The authors of GS-WGAN (Chen et al., 2020) adopt Wasserstein GAN (WGAN, Arjovsky et al., 2017) and propose using Wasserstein-1 loss for training. They

show that such an approach can distort gradient information more precisely; thus GS-WGANs generate more informative samples. The authors of DataLens (Wang et al., 2021) leverage GANs to reduce the gradient noise using gradient compression. In a similar direction, to improve information capture from the gradient, the authors of DPGEN[1] (Chen et al., 2022) deploy an energy-guided network. They train on sanitized data to indicate the direction of the actual data distribution via the Langevin Markov chain Monte Carlo sampling method. However, since all of these techniques rely on GANs, they are susceptible to training instability of GANs. To address the issues with GAN-based methods, the authors of DP-MERF (Harder et al., 2021) synthesize images by taking advantage of random feature representations of kernel mean embeddings, while the authors of P3GM (Takagi et al., 2021) abandon GANs in favor of a variant of a private variational autoencoder.

So far, we have discussed generative DP techniques; next, we discuss discriminative DP techniques. Private Aggregation of Teacher Ensembles (PATE, Papernot et al., 2017; 2018) divides the training data into disjoint sets and assigns them to multiple classifiers (teachers). The teachers are queried with either public or GAN-generated images to obtain the corresponding soft labels (Long et al., 2021; Jordon et al., 2019). The student network is trained using public or GAN-generated images and their soft labels using knowledge transfer (Hinton et al., 2015). To maintain the privacy of multiple teacher models, the soft labels of all the teachers are aggregated, and noise is added before they are released for training the student. When PATE employs a GAN for image generation, it encounters the aforementioned challenges related to GANs. When utilizing public images, it struggles to address situations where public and private images are not aligned. Thus, to address the challenges of generative and discriminative DP data release techniques, we propose a new framework DP-ImgSyn.

## 3 DP-ImgSyn: Differentially Private Image Synthesis

This section introduces our approach's specifics, as Figure 3 illustrates. Our method consists of three steps: 1) train a teacher model with a DP training algorithm on the private images, 2) perform optimization on the public dataset to align it to the private dataset and 3) generate soft labels for the aligned synthetic public dataset. Finally, the synthetic images and their corresponding soft labels are publicly released to train a student network.

### 3.1 DP Teacher Model Training

The first step of our method involves training the teacher model utilizing a DP training algorithm. Note that the teacher model is not trained with standard SGD but with a DP-based training algorithm to ensure privacy. We select the Differential Private-Stochastic Gradient Descent (DP-SGD) (Abadi et al., 2016) as the DP training algorithm. The DP-SGD algorithmic technique trains a neural model with $(\epsilon, \delta)$-DP guarantees given a privacy budget. Similar to standard SGD, the algorithm converges in multiple training steps. At each training step $t$, DP-SGD computes the gradient $g_t(x)$ of the loss function with respect to the model parameters for a training image $x$. Then, it clips each gradient vector $g$ to have a maximum $l_2$ norm of $C$. That is, the gradient vector $g$ is replaced by $g/max(1, ||g||_2/C)$. The clipping ensures that if $||g||_2 \leq C$ then $g$ is preserved, whereas if $||g||_2 > C$, it gets scaled down to be of norm $C$. Thus, the contribution of each data point to the batch gradient is bound by a constant $C$. Noise is added to the gradient $g_t(x) + \mathcal{N}(0, \sigma^2 C^2 I)$ and the descent step $\theta_{t+1} = \theta_t - \eta_t g_t$ is performed, with $\eta_t$ learning rate. After $T$ iterations, it outputs the $(\epsilon_{train}, \delta)$-DP teacher model.

The next step's synthesis (a.k.a dataset alignment) requires the teacher model's batch statistics. These are obtained from the batch norm layer of the teacher model. However, batch norm layers cannot be used for DP training. The batch norm computes the mean over multiple training data points. Thus, per sample gradient cannot be obtained during training. The gradient norm cannot be bound without per sample gradient, so we cannot provide DP guarantees. To address this issue, we propose the following DP-guaranteed approach to obtain batch statistics. First, we use group norm layers instead of batch norm

---

[1]Please note, DPGEN (Chen et al., 2022) privacy guarantees are compromised due to conceptual errors reported in Dockhorn et al. (2022).

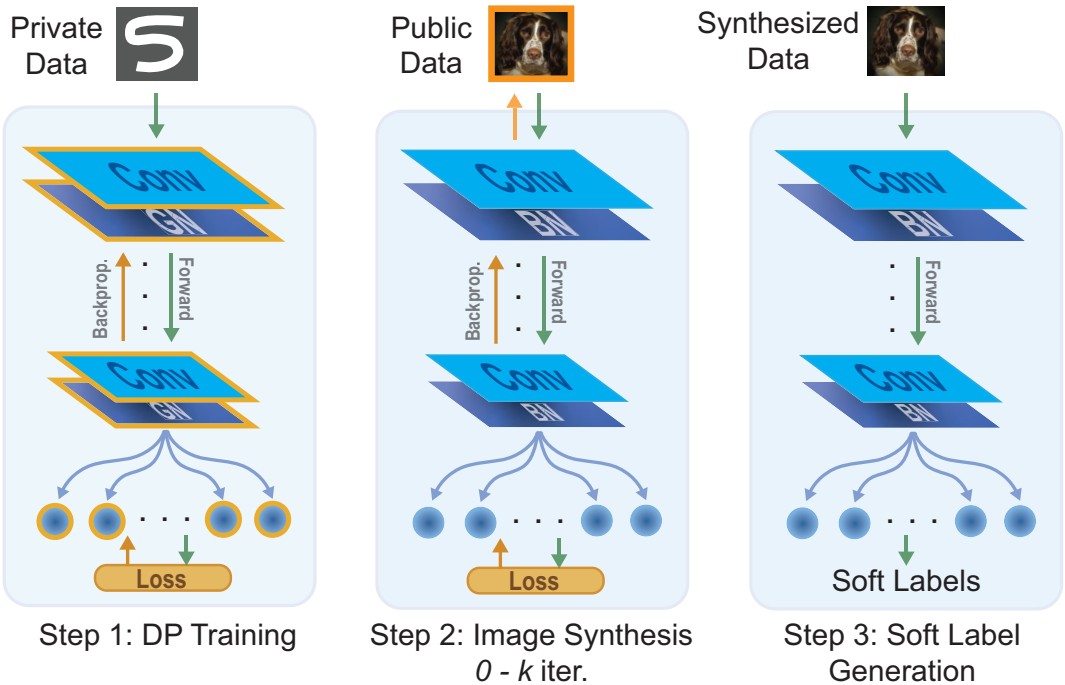

Figure 3: Overview of DP-ImgSyn: (1) Train a teacher model using a DP-training scheme. Capture the batch statistics of the model on the private dataset using the proposed DP guaranteed technique. (2) Perform the proposed public dataset alignment to obtain a better-aligned synthetic public dataset. (3) Generate soft labels and release the synthesized images and soft labels as the sanitized dataset.

layers. Next, we train the teacher model using the previously described DP-SGD. Once trained, we capture the input to all group norm layers, $i_{gn}$. We clip the input $i_{gn}$ to have a maximum $l_2$ norm of $C$ and add noise, $\hat{i}_{gn} = i_{gn} + \mathcal{N}(0, \sigma^2 C^2 I)$. The noisy input $\hat{i}_{gn}$ is used to calculate the batch statistics of the private set. This process has the same DP guarantee as the Gaussian Mechanism (Dwork et al., 2014) since it uses the Gaussian DP mechanism described in Dwork et al. (2014). Note that obtaining the batch statistics this way consumes some of the privacy budget allocated for training. Thus, when we report our results, the privacy budget $\epsilon$ is for the combined training and statistics capture process. To be more specific the reported budget $\epsilon = composition(\epsilon_{train}, \epsilon_{batch-stats})$. We use the accountant implementation from Gopi et al. (2021) to calculate the upper bounds on privacy. To ensure correctness, we also verify the upper bounds on privacy using both Opacus (Yousefpour et al., 2021) and Gopi et al. (2021) implementations.

### 3.2 DP Image Synthesis

Image synthesis (a.k.a dataset alignment) optimizes the public dataset to align with the private dataset to ensure that the synthesized (or aligned) images will have good distillation performance. To perform image synthesis, we collect the layerwise batch statistics of the private dataset using the DP-guaranteed technique described in Section 3.1. The batch statistics consists of the mean $\mu = [\mu_1, \cdots, \mu_L]$ and the variance $\sigma = [\sigma_1, \cdots, \sigma_L]$ from the all the $L$ layers of the teacher model $\mathcal{M}$. Let $\hat{x}$ denote a batch of synthetic images and $x_P$ denote a batch of data sampled from a public dataset $D_p$. Algorithm 1 summarizes the DP image synthesis process. We initialize $\hat{x}$ with $x_P$, this corresponds to Line 1 in Algorithm 1. For each data point in $\hat{x}$, we assign a target label $y$ (Line 2 in Algorithm 1). The target labels are uniformly distributed over all the classes, that is the assignment is such that we have the same number of images for each class. This step of generating the labels is independent of the private dataset to ensure privacy. For the exact implementation please see Appendix A.1.3.

---

**Algorithm 1:** DP Image Synthesis

---

**Input:** DP-trained teacher model $\mathcal{M}$; $k$ number of optimization iterations; synthesis learning rate $\gamma_{syn}$;
        batch statistics $\mu, \sigma$ for the private set; batch of public images $x_P$

**Output:** $\hat{x}$, one batch of aligned synthetic images

**1** $\hat{x} \leftarrow x_P$

**2** $y \leftarrow$ Target labels for batch $\hat{x}$, ==uniformly distributed over all the classes==

**3 for** $i = 1, 2, ..., k$ **do**

**4**     $\mu(\hat{x}), \sigma(\hat{x}) \leftarrow \mathcal{M}(\hat{x})$

**5**     $\mathcal{R} \leftarrow \mathcal{R}_{total}(\hat{x}, y, \mu, \sigma, \mu(\hat{x}), \sigma(\hat{x}))$ ;                     `// Compute the loss from Equation 5`

**6**     $\nabla_{\hat{x}}\mathcal{R} \leftarrow$ Backward pass

**7**     Update $\hat{x} \leftarrow \hat{x} - \gamma_{syn}\nabla_{\hat{x}}\mathcal{R}$

**8 return** $\hat{x}$

---

The next step is to perform $k$ iterations of optimization corresponding to Lines 3 - 7. These $k$ iterations optimize $\hat{x}$ to align with the private set. Each iteration consists of a forward pass of $\hat{x}$ through the DP-trained teacher model $\mathcal{M}$ (Line 4 in Algorithm 1). The forward pass is used to obtain the layer-wise batch statistics for $\hat{x}$, $(\mu(\hat{x}), \sigma(\hat{x}))$. The batch statistics and the generated label $y$ are used to calculate the loss described in Equation 5 (Line 5 in Algorithm 1). The gradient of the loss $\mathcal{R}$ with respect to $\hat{x}$ ($\nabla_{\hat{x}}\mathcal{R}$) is calculated using back-propagation and is used to update the image $\hat{x}$ (Lines 6 - 7 in Algorithm 1). At the end of $k$ update steps, we have synthesized one batch of aligned images $\hat{x}$. Since the teacher model is DP-trained, and the private dataset batch statistics are obtained with a DP guarantee, image synthesis is also DP-guaranteed.

The loss used to guide the optimization is critical. The total loss $\mathcal{R}_{total}$ consists of the following terms: feature loss $\mathcal{R}_{feature}$, classification loss $\mathcal{R}_{classif}$, total variance loss $\mathcal{R}_{tv}$ and $l_2$ norm loss $\mathcal{R}_{l_2}$. The sum of total variance loss $\mathcal{R}_{tv}$ and the $l_2$ norm loss $\mathcal{R}_{l_2}$ are referred to as prior loss. Next, we define each of these losses. The feature loss $\mathcal{R}_{feature}$ computes the distance between the batch statistics of the private dataset and the synthetic set $\hat{x}$ and is given by the following equation:

$$\mathcal{R}_{feature}(\mu, \sigma, \mu(\hat{x}), \sigma(\hat{x})) = \sum_{l=1}^{L} ||\mu_l(\hat{x}) - \mu_l||_2^2 + ||\sigma_l(\hat{x}) - \sigma_l)||_2^2 \tag{1}$$

Where $\hat{x}$ is the synthesized-aligned image, $\mu_l(\hat{x})$ and $\sigma_l(\hat{x})$ are the batch-wise mean and variance estimates of feature maps corresponding to the $l^{th}$ layer when $\hat{x}$ is fed to $\mathcal{M}$, and $\mu_l$ and $\sigma_l$ are the $l^{th}$ layer batch statistics obtained from the private dataset described in Section 3.1.

The classification loss $\mathcal{R}_{classif}$ is the cross-entropy loss between the teacher output and the target label $y$ and is defined as:

$$\mathcal{R}_{classif}(\hat{x}, y) = \mathcal{L}(p_{\mathcal{M}}(\hat{x}), y) \tag{2}$$

where $\mathcal{L}$ is the cross-entropy loss, $p_{\mathcal{M}}(\hat{x})$ is the output of the teacher model $\mathcal{M}$ when $\hat{x}$ is fed as input, and $y$ is the target label.

The total variance loss $\mathcal{R}_{tv}$ ensures no sharp transitions in the synthetic image and restricts the adjacent pixels to have similar values. It is defined as:

$$\mathcal{R}_{tv}(\hat{x}) = \sum_{i,j}((\hat{x}_{i,j+1} - \hat{x}_{i,j})^2 + (\hat{x}_{i+1,j} - \hat{x}_{i,j})^2)^{\frac{1}{2}} \tag{3}$$

==The $l_2$ norm loss is employed to encourage the image range to remain within a target interval rather than diverging.== The $l_2$ norm loss $\mathcal{R}_{l_2}$ for the $\hat{x}$ is defined as:

$$\mathcal{R}_{l_2}(\hat{x}) = ||\hat{x}||_2^2 \tag{4}$$

The total loss is the sum of the aforementioned losses:

$$\mathcal{R}_{total}(\hat{x}, y, \mu, \sigma) = \alpha_f \mathcal{R}_{feature}(\hat{x}, \mu, \sigma) + \alpha_c \mathcal{R}_{classif}(\hat{x}, y) + \alpha_{tv} \mathcal{R}_{tv}(\hat{x}) + \alpha_{l_2} \mathcal{R}_{l_2}(\hat{x}) \tag{5}$$

Each loss term is multiplied by a corresponding scaling factor $\alpha_f, \alpha_c, \alpha_{tv}, \alpha_{l_2}$. The teacher model is not updated during back-propagation, and only $\hat{x}$ is optimized. After $k$ iterations, we obtain the synthetic DP images $\hat{x}$.

### 3.3 DP Image Release

After synthesizing the images, we obtain their corresponding soft labels. The synthetic images are fed to the DP-trained teacher model $\mathcal{M}$, and the corresponding soft labels $\hat{y} = p_{\mathcal{M}}(\hat{x})$ are recorded. Because the teacher model is trained with DP-SGD, querying the teacher to obtain the soft labels does not impose privacy risk. The synthetic images $\hat{x}$, along with their soft labels $\hat{y}$, are publicly released. The student model $\mathcal{S}$ is trained on the synthetic images $\hat{x}$ and their corresponding soft labels $\hat{y}$ using KL-divergence:

$$\min_{\theta} \sum_{x \in \mathcal{X}^s} KL(\hat{y}, p_{\mathcal{S}}(\hat{x})/T) \tag{6}$$

$KL$ refers to the Kullback-Leibler divergence, $p_{\mathcal{S}}(\hat{x})$ is the output (soft labels) of the student model when the synthetic image $\hat{x}$ is given as input. $T$ is a scaling temperature value.

## 4 Experimental Evaluation

### 4.1 Experimental Setup

To evaluate our proposal, we use the same vision datasets as previous works; specifically, we use MNIST (Le-Cun et al., 1998), FashionMNIST (Xiao et al., 2017), CIFAR-10 (Krizhevsky et al., 2009), ImageNette (ima, 2018), CelebA-Hair (Long et al., 2021; Liu et al., 2015), CelebA-Gender (Long et al., 2021; Liu et al., 2015), TinyImageNet (Li et al., 2015), Places365 (Zhou et al., 2017), LSUN (Yu et al., 2015), and Textures (Cimpoi et al., 2014). We use the following networks architectures: ResNet18 (He et al., 2016), ResNet34 (He et al., 2016), VGG11 (Simonyan & Zisserman, 2014), MobileNetV2 (Sandler et al., 2018), and ShuffleNetV2 (Ma et al., 2018). All the models were trained till convergence or privacy budget exhaustion. The detailed hyper-parameter settings for image synthesis and model training, computational resources, and dataset statistics are reported in the Appendix. We perform the experiments described in the following sections to evaluate our proposal thoroughly.

### 4.2 Number of Optimization Iterations $k$

This section studies the effect of the number of optimization iterations on image synthesis and performance.

**Experiment** We train a ResNet34 till convergence on the private dataset CIFAR-10. Next, we perform the alignment optimization detailed in Section 3.2 for various numbers of iterations (i.e., $k$ in DP-ImgSyn Algorithm 1) ranging from 0-100. For the alignment, we use TinyImageNet as the public dataset. The DP-ImgSyn generated dataset is used to train a ResNet18 student model. The student model accuracy and losses versus the number of optimization iterations $k$ are reported. We use privacy budget $\epsilon = \infty$ to isolate all variables. The results are visualized in Figure 4.

**Results** Figure 4a illustrates the accuracy of the student model and the total loss $\mathcal{R}_{total}$ versus the number of optimization iterations $k$. This is visualized with two y-axes: the left axis for accuracy and the right axis for the total loss $\mathcal{R}_{total}$. Note that the student model accuracy peaks around $k = 20$ iterations. Continuing the optimization by increasing $k$ reduces student model performance. The plot in Figure 4b explains the reason for this behavior. Figure 4b plots the accuracy of the student model and the total variance loss $\mathcal{R}_{tv}$ versus the number of optimization iterations $k$. This is visualized with two y-axes, left for accuracy and the

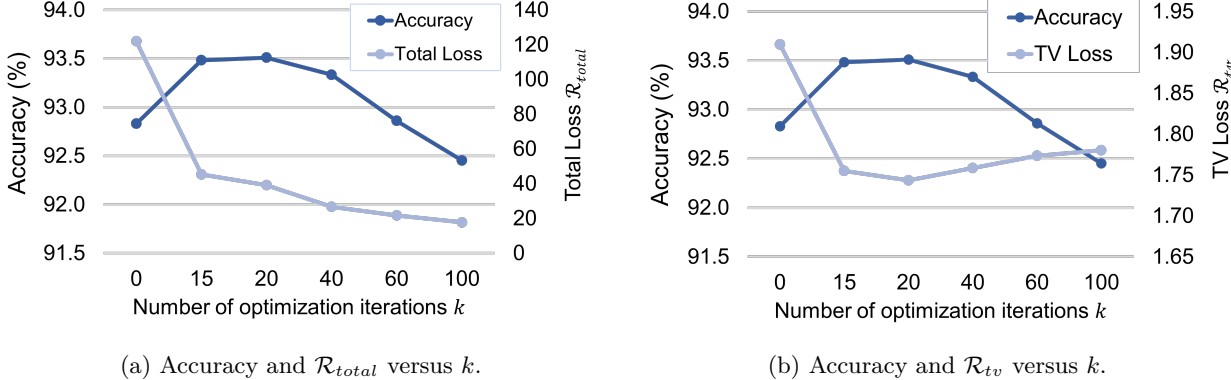

(a) Accuracy and $\mathcal{R}_{total}$ versus $k$.

(b) Accuracy and $\mathcal{R}_{tv}$ versus $k$.

Figure 4: Plots for student model accuracy, total loss $\mathcal{R}_{total}$ and total variance loss $\mathcal{R}_{tv}$ versus the number of optimization iterations $k$. The left axis is for accuracy, and the right is for loss $\mathcal{R}_{total}$ and $\mathcal{R}_{tv}$, respectively. Results suggest early stopping is necessary to optimize accuracy. For this plot, DP-ImgSyn is initialized with the TinyImageNet dataset, and the ResNet34 is the teacher model.

right for total variance loss $\mathcal{R}_{tv}$. The loss $\mathcal{R}_{tv}$ expresses the image prior. The image prior ensures no sharp transitions in the synthetic image. It is used in literature (Dosovitskiy & Brox, 2016; Mahendran & Vedaldi, 2015; Nguyen et al., 2015; Simonyan et al., 2014) as a proxy for how natural synthesized images are. This plot suggests that the images become more artificial or synthesized as we optimize past a certain threshold. It is observed that around 20 iterations, the $\mathcal{R}_{tv}$ loss starts increasing, and the accuracy starts decreasing.

**Experiment Conclusion** Minimizing $\mathcal{R}_{total}$ does not guarantee optimal image prior ($\sim \mathcal{R}_{tv}$). However, $\mathcal{R}_{tv}$ significantly impacts student model accuracy. Thus, early stopping is necessary to optimize student model performance.

### 4.3   Privacy and Performance

We evaluate the performance of DP-ImgSyn with 0 iterations and $k$ iterations on various vision datasets as public and private datasets.

**Experiment** First, we select a private dataset. We train a teacher ResNet18 model on the private dataset using DP-SGD and capture batch statistics as described in Section 3.1. This results in a ResNet18 teacher model with batch statistics having a privacy budget of $\epsilon$. Next, we select a public dataset and synthesize the released/sanitized dataset described in Sections 3.2 and 3.3. A student ResNet18 model is trained on the synthesized dataset with various public datasets as initialization. We report the accuracy results for privacy budgets $\epsilon \in \{1, 10\}$ on MNIST, FashionMNIST, CelebA-Hair, and CelebA-Gender as private datasets. We select public datasets whose size is the same or exceeds the size of the private dataset. This ensures that the synthesized dataset has the same size as the private dataset. When the public dataset is larger than the private set, we randomly sample from the public set to obtain a subset of the same size as the private set. This public subset is used for the synthesis. The number of classes of the public dataset is not required to be the same as the private set, because we use the soft labels of the teacher model. Note that, there is *no* one-to-one correspondence between public and private set images. See Appendix A.1.4 for an ablation study of the various loss terms.

**Results** Table 1 reports the accuracy of the ResNet18 DP-teacher model trained on various datasets. Table 2 summarizes the accuracy results for MNIST, FashionMNIST, CelebA-Hair, and CelebA-Gender as private datasets and TinyImageNet, Places365, FashionMNIST, MNIST, and LSUN as public datasets. Table 2 reports the performance of the student model when performing 0 iterations, reported as DP-ImgSyn(0). The student model is trained on the DP-ImgSyn generated images and the test set is the private test set. We report the performance with early stopping at $k_{exp}$ iterations as DP-ImgSyn($k_{exp}$) and the optimal iterations to stop as $k_{opt}$.

Table 1: Accuracy for ResNet18 DP-teacher model with $\epsilon \in \{1, 10\}$ for MNIST, FashionMNIST, CelebA-Hair, and CelebA-Gender.

|  | MNIST | FashionMNIST | CelebA-Hair | CelebA-Gender |
|---|---|---|---|---|
| $\epsilon = 1$ | 86.87% | 76.27% | 79.95% | 91.02% |
| $\epsilon = 10$ | 96.30% | 81.88% | 81.74% | 92.35% |

Table 2: Comparative Table with $\epsilon \in \{1, 10\}$ for MNIST, FashionMNIST, CelebA-Hair, and CelebA-Gender as private datasets using TinyImageNet, Places365, FashionMNIST, MNIST, and LSUN as public datasets. Results are mean $\pm$ std over three different seeds. The models are trained on the synthetic images generated by DP-ImgSyn (training set) and evaluated on the test set of the private dataset (testing set).

| Private Dataset | $\epsilon$ | Public Dataset | DP-ImgSyn(0) | DP-ImgSyn($k_{exp}$) | $k_{exp}$ | $k_{opt}$ |
|---|---|---|---|---|---|---|
| MNIST | $\epsilon = 1$ | TinyImageNet | $85.83 \pm 0.13$ | $\mathbf{85.98 \pm 0.06}$ | 10 | 10 |
|  |  | Places365 | $85.00 \pm 0.30$ | $\mathbf{86.01 \pm 0.22}$ | 10 | 10 |
|  |  | FashionMNIST | $85.56 \pm 0.32$ | $\mathbf{86.24 \pm 0.03}$ | 10 | 10 |
|  | $\epsilon = 10$ | TinyImageNet | $92.97 \pm 0.65$ | $\mathbf{94.03 \pm 0.64}$ | 10 | 10 |
|  |  | Places365 | $92.63 \pm 0.23$ | $\mathbf{93.74 \pm 0.18}$ | 10 | 10 |
|  |  | FashionMNIST | $93.61 \pm 0.37$ | $\mathbf{93.90 \pm 0.30}$ | 10 | 10 |
| FashionMNIST | $\epsilon = 1$ | TinyImageNet | $74.99 \pm 0.21$ | $74.93 \pm 0.02$ | 1 | 0 |
|  |  | Places365 | $75.08 \pm 0.15$ | $74.94 \pm 0.20$ | 1 | 0 |
|  |  | MNIST | $51.58 \pm 2.28$ | $\mathbf{68.38 \pm 0.34}$ | 10 | 10 |
|  | $\epsilon = 10$ | TinyImageNet | $79.04 \pm 0.04$ | $78.71 \pm 0.20$ | 1 | 0 |
|  |  | Places365 | $78.73 \pm 0.14$ | $\mathbf{78.80 \pm 0.05}$ | 1 | 1 |
|  |  | MNIST | $54.78 \pm 1.15$ | $\mathbf{71.51 \pm 1.49}$ | 10 | 10 |
| CelebA-Hair | $\epsilon = 1$ | LSUN | $79.89 \pm 0.08$ | $79.42 \pm 0.14$ | 1 | 0 |
|  |  | Places365 | $79.91 \pm 0.08$ | $79.50 \pm 0.15$ | 1 | 0 |
|  | $\epsilon = 10$ | LSUN | $81.31 \pm 0.04$ | $79.28 \pm 0.20$ | 1 | 0 |
|  |  | Places365 | $81.33 \pm 0.12$ | $78.73 \pm 0.44$ | 1 | 0 |
| CelebA-Gender | $\epsilon = 1$ | LSUN | $89.91 \pm 0.15$ | $89.17 \pm 0.26$ | 1 | 0 |
|  |  | Places365 | $90.06 \pm 0.05$ | $89.03 \pm 0.19$ | 1 | 0 |
|  | $\epsilon = 10$ | LSUN | $90.99 \pm 0.16$ | $89.90 \pm 0.26$ | 1 | 0 |
|  |  | Places365 | $91.22 \pm 0.04$ | $89.27 \pm 0.99$ | 1 | 0 |

**Experiment Conclusion** From Table 2, we observe that, on average, when the private datasets are aligned over various public datasets, DP-ImgSyn(k) performs similar to DP-ImgSyn(0). For some datasets, we observe $k_{opt} = 0$, i.e., the proposed alignment process does not improve performance much. However, when the datasets are misaligned, like in the case of FashionMNIST and MNIST, we see DP-ImgSyn(k) performs significantly better ($\approx$**17%** improvement in student model accuracy, for FashionMNIST private dataset with MNIST public dataset initialization for $\epsilon \in \{1, 10\}$).

Table 3: Comparison Table with state-of-the-art techniques for $\epsilon \in \{1, 10\}$ for MNIST, FashionMNIST, CelebA-Hair, and CelebA-Gender. Results for DP-ImgSyn are mean over three different seeds. The best-performing framework is highlighted in bold, and the second-best is underlined. DP-GAN refers to Xie et al. (2018), DP-MERF refers to Harder et al. (2021), P3GM refers to Takagi et al. (2021), DataLens refers to Wang et al. (2021) and G-PATE refers to Long et al. (2021).

| Dataset | $\epsilon$ | DP-GAN | DP-MERF | P3GM | DataLens | G-PATE | DP-ImgSyn (ours) |
|---|---|---|---|---|---|---|---|
| MNIST | 1 | 40.36% | 63.67% | 73.69% | 71.23% | 58.80% | **86.24%** |
| | 10 | 80.11% | 67.38% | 79.81% | 80.88% | 80.92% | **94.03%** |
| FashionMNIST | 1 | 10.53% | 58.62% | 72.23% | 64.78% | 58.12% | **75.08%** |
| | 10 | 60.98% | 61.62% | 74.80% | 70.61% | 69.34% | **79.04%** |
| CelebA-Hair | 1 | 34.47% | 44.13% | 45.32% | 60.61% | 49.85% | **79.91%** |
| | 10 | 39.20% | 52.25% | 44.89% | 62.24% | 62.17% | **81.33%** |
| CelebA-Gender | 1 | 53.30% | 59.36% | 56.73% | 69.96% | 67.02% | **90.06%** |
| | 10 | 52.11% | 60.82% | 58.84% | 72.87% | 68.97% | **91.22%** |

## 4.4 Comparison with other Techniques

**Experiment** This section compares our proposal with previous approaches: DP-GAN (Xie et al., 2018), DP-MERF (Harder et al., 2021), P3GM (Takagi et al., 2021), DataLens (Wang et al., 2021) and G-PATE (Jordon et al., 2019). For a fair comparison, we compare with techniques that use similar DP-training schemes (i.e., variants of DP-SGD). Note, that we exclude comparison with DPGEN (Chen et al., 2022) because privacy guarantees are compromised due to errors as reported in Dockhorn et al. (2022). The results for other techniques are the best accuracy results reported in prior publications. For our results, we report our best performance results from Table 2. Specifically, for $\epsilon = 1$, we use as public dataset initialization FashionMNIST, MNIST, Places, and Places for the private datasets MNIST, FashionMNIST, CelebA-Hair, and CelebA-Gender, respectively. For $\epsilon = 10$, we use as public dataset initialization TinyImageNet, TinyImageNet, LSUN, and Places for the private datasets MNIST, FashionMNIST, CelebA-Hair, and CelebA-Gender, respectively.

**Results** Table 3 compares the performance of the proposed technique with state-of-the-art techniques.

**Experiment Conclusion** We observe that our proposed method significantly outperforms generative techniques that use similar DP training schemes (up to $\approx$ **20%**, for both CelebA-Hair and CelebA-Gender for $\epsilon = 1$).

## 4.5 Beyond Generative Methods

This section presents results on higher resolution (224 x 224) and more varied datasets with which generative techniques often have difficulty.

**Experiment** We sanitized ImageNette and CIFAR-10 using the proposed DP-ImgSyn technique. For ImageNette, we use a resolution of $224 \times 224$ with $\epsilon = 105$, initialized with Textures as public dataset; for CIFAR-10, we select $\epsilon = 10$, initialized with TinyImageNet as public dataset. Note that generative methods do not report results on these datasets.

**Results** A ResNet18 student model trained on the DP-ImgSyn generated dataset for ImageNette ($224 \times 224$) achieved **39.38%** accuracy on the test set, while the teacher model achieves 43.26% accuracy. Similarly, for CIFAR-10, a ResNet18 student trained on DP-ImgSyn generated dataset achieved **45.66%** accuracy on the test set, while the teacher model achieves 42.95% accuracy.

Table 4: Comparative Table with $\epsilon \in \{1, 10\}$ for MNIST, and CelebA-Gender as private datasets using TinyImageNet, Places365, FashionMNIST, and LSUN and Places365 as public datasets respectively. Results are mean $\pm$ std over three different seeds. The models are trained on the synthetic images generated by DP-ImgSyn (training set) using a ResNet18 as teacher model. The student model architecture is VGG11, MobileNetV2, ShuffleNetV2, and ResNet18. The student models are evaluated on the test set of the private dataset (testing set).

| Private Dataset | $\epsilon$ | Public Dataset | Student Model Architecture | | | |
|---|---|---|---|---|---|---|
| | | | VGG11 | MobileNetV2 | ShuffleNetV2 | ResNet18 |
| MNIST | $\epsilon = 1$ | TinyImageNet | $85.56 \pm 0.28$ | $84.34 \pm 0.70$ | $85.15 \pm 0.16$ | $85.98 \pm 0.06$ |
| | | Places365 | $85.65 \pm 0.48$ | $84.02 \pm 0.33$ | $85.28 \pm 0.22$ | $86.01 \pm 0.22$ |
| | | FashionMNIST | $86.37 \pm 0.42$ | $84.95 \pm 0.28$ | $85.91 \pm 0.06$ | $86.24 \pm 0.03$ |
| | $\epsilon = 10$ | TinyImageNet | $94.57 \pm 0.25$ | $92.08 \pm 0.62$ | $94.04 \pm 0.09$ | $94.03 \pm 0.64$ |
| | | Places365 | $93.91 \pm 0.26$ | $91.75 \pm 0.74$ | $93.22 \pm 0.24$ | $93.74 \pm 0.18$ |
| | | FashionMNIST | $94.70 \pm 0.18$ | $92.66 \pm 0.95$ | $92.91 \pm 0.91$ | $93.90 \pm 0.30$ |
| CelebA-Gender | $\epsilon = 1$ | LSUN | $88.32 \pm 0.32$ | $88.31 \pm 0.30$ | $89.44 \pm 0.17$ | $89.17 \pm 0.26$ |
| | | Places365 | $88.61 \pm 0.38$ | $88.49 \pm 0.14$ | $89.16 \pm 0.44$ | $89.03 \pm 0.19$ |
| | $\epsilon = 10$ | LSUN | $89.36 \pm 0.55$ | $88.85 \pm 0.25$ | $89.54 \pm 0.28$ | $89.90 \pm 0.26$ |
| | | Places365 | $89.40 \pm 0.50$ | $87.85 \pm 0.26$ | $89.02 \pm 0.02$ | $89.27 \pm 0.99$ |

**Experiment Conclusion** The ImageNette results suggest that DP-ImgSyn is not limited by the image resolution, unlike existing generative techniques. ImageNette and CIFAR-10 results indicate that the DP-ImgSyn technique is better suited to more complex and higher-resolution datasets when compared to generative techniques.

### 4.6 Generalization to Other Network Architectures

This section presents results with various network architectures for the student network to evaluate whether our released dataset can be used to train any network architecture.

**Experiment** We use DP-ImgSyn to generate the synthetic images using a ResNet18 as the teacher model. The student model architecture is VGG11 (Simonyan & Zisserman, 2014), MobileNetV2 (Sandler et al., 2018), ShuffleNetV2 (Ma et al., 2018), and ResNet18 (He et al., 2016). For $\epsilon = 1, 10$ we use MNIST and CelebA Gender as the private dataset initialized with TinyImageNet, Places365, FashionMNIST, and LSUN, Places365 respectively.

**Results** Table 4 summarizes the results across the various student architectures. We observe that the accuracy is similar across the architectures (VGG11, MobileNetV2, ShuffleNetV2), and similar to ResNet18 which has the same architecture as the teacher model.

**Experiment Conclusion** The outcome of this experiment is that the synthetic images can be used to train any student network architecture. Thus, DP-ImgSyn is not restricted by the teacher model architecture.

## 5 Conclusion

Deep neural models are state-of-the-art solutions for various tasks in multiple domains, but they require lots of available data for training. Certain applications in the real world require the classification of visually disturbing and sensitive images (for example content moderation images) and releasing such images becomes challenging. This paper presents a discriminative approach (DP-ImgSyn) for releasing images that are visually dissimilar to the private images and have DP guarantees. The proposed technique is not a new

DP mechanism but is rather a new approach to visual privacy for sensitive datasets which leverages dataset alignment to obfuscate sensitive data in public datasets. DP-ImgSyn offers a new technique to align public and private datasets. This alignment/synthesis process improves the performance of even highly misaligned public-private dataset pairs. We observe $\approx$ **17%** improvement in the performance of highly misaligned datasets. We also show that the non-generative DP-ImgSyn approach significantly outperforms (up to $\approx$ **20%**) generative techniques using similar DP-training schemes. Moreover, we present results on higher resolution (224 x 224) and more varied datasets with which generative techniques often have difficulty. Further, synthesized images generated by DP-ImgSyn are visually dissimilar to the private dataset, as DP-ImgSyn injects information from the private dataset into the public dataset with a DP guarantee. Our findings suggest discriminative (a.k.a non-generative) approaches might better suit dataset sanitization. However, further research is needed to identify the limits of discriminative and generative techniques.

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

# A  Appendix

## A.1  Training Details

DP-ImgSyn implementation uses the Pytorch (Paszke et al., 2019) framework, and the experiments were conducted on NVIDIA GeForce GTX 1080 Ti with 11 GB of memory with the Ubuntu operating system.

### A.1.1  DP Teacher Model Hyperparameters

For the DP statistics capture for the teacher model (Section 3.1 from the main paper), we used the hyperparameters reported in Table 5. The $\epsilon$ denotes the privacy budget, $\eta_{tr}$ is the number of training epochs, $\Omega_{tr}$ is the batch size used for DP-SGD training, $\gamma_{tr}$ is learning rate used for training, $C$ denotes the maximum norm limit for the gradient vector $g$ ($g/max(1, ||g||_2/C)$), $\sigma$ controls the amount of noise added to $g$ ($g + \mathcal{N}(0, \sigma^2 C^2 I)$), $\eta_{bn}$ is the number of epochs used for capturing batch statistics, and $\Omega_{bn}$ is the batch size used for capturing batch statistics in Table 5. The accuracy results of the corresponding teacher models are reported in Table 1 in the main paper.

Table 5: Hyperparameters used for DP training and batch statistics capture experiments on the vision datasets: MNIST, FashionMNIST, CelebA-Hair, CelebA-Gender, CIFAR-10, and ImageNette. The $\epsilon$ denotes the privacy budget, $\eta_{tr}$ is the number of training epochs, $\Omega_{tr}$ is the batch size used for DP-SGD training, $\gamma_{tr}$ is learning rate used for training, $C$ denotes the maximum norm limit for the gradient vector $g$ ($g/max(1, ||g||_2/C)$), $\sigma$ controls the amount of noise added to $g$, $\eta_{bn}$ is the number of epochs used for capturing batch statistics, and $\Omega_{bn}$ is the batch size used for capturing batch statistics

| Dataset | $\epsilon$ | $\eta_{tr}$ | $\Omega_{tr}$ | $\gamma_{tr}$ | $C$ | $\sigma$ | $\eta_{bn}$ | $\Omega_{bn}$ |
|---|---|---|---|---|---|---|---|---|
| MNIST | 1 | 4 | 128 | 0.01 | 1.0 | 0.8 | 2 | 64 |
| | 10 | 14 | 128 | 0.01 | 1.0 | 0.5 | 2 | 64 |
| FashionMNIST | 1 | 30 | 50 | 0.01 | 1.2 | 1 | 2 | 64 |
| | 10 | 20 | 128 | 0.01 | 1.2 | 0.5 | 2 | 64 |
| CelebA Hair | 1 | 18 | 128 | 0.001 | 1.0 | 0.8 | 5 | 128 |
| | 10 | 22 | 128 | 0.001 | 1.0 | 0.45 | 3 | 128 |
| CelebA Gender | 1 | 18 | 128 | 0.001 | 1.0 | 0.8 | 4 | 128 |
| | 10 | 22 | 128 | 0.001 | 1.0 | 0.5 | 4 | 128 |
| CIFAR-10 | 10 | 12 | 128 | 0.001 | 1.0 | 0.5 | 5 | 128 |
| ImageNette | 105 | 57 | 8 | 0.001 | 1.0 | 0.3 | 3 | 8 |

### A.1.2 DP Image Synthesis Hyperparameters

For the DP Image Synthesis (described in Section 3.2, and Algorithm 1 in the main paper), we use the Adam optimizer (Kingma & Ba, 2014) with synthesis learning rate $\gamma_{syn} = 0.1$, betas $\beta_1 = 0.5$, $\beta_2 = 0.99$. For MNIST and FashionMNIST, we use a batch size of 80; for CelebA-Hair and CelebA-Gender, we use a batch size of 60. The main paper in Table 2 reports the number of optimization iterations for each dataset and privacy budget.

The total loss optimized during image synthesis is the summation of the following losses: $\mathcal{R}_{feature}$, $\mathcal{R}_{classif}$, $\mathcal{R}_{tv}$, and $\mathcal{R}_{l_2}$ (Equation 5 in the main paper). The scaling coefficients corresponding to each of these losses are denoted as $\alpha_f$, $\alpha_c$, $\alpha_{tv}$, and $\alpha_{l_2}$, respectively. Table 7 reports the values of the scaling coefficients used in our simulations. Furthermore, we perform an ablation study on the scaling coefficients that control the total loss on MNIST as private dataset for epsilon 10, and TinyImageNet as the public dataset initialization. Specifically, we search the values for each loss scaling factor while keeping the remaining scaling factors constant. Table 6 summarizes our results. We observe that our method is robust to the scaling factor hyperparameter selections. Then, we repeated the experiment using the same setup (MNIST as the private dataset with epsilon set to 10), but with Gaussian noise as initialization. The findings in Table 6 consistently affirm our earlier observation, highlighting the robustness of our method to different scaling factor hyperparameter selections.

### A.1.3 DP-ImgSyn Label Generation Implementation

Here we provide the PyTorch code that we use for generating the targets for DP-ImgSyn, given the number of classes and batch size. Since the label generation algorithm is independent of the data, there is no privacy leakage.

```python
def generate_labels(num_classes, batch_size):
    x = torch.arange(num_classes)
    targets = torch.squeeze(x.repeat(1, int(batch_size/num_classes)))
    return targets
```

Table 6: Ablation study on the scaling coefficients $\alpha_f$, $\alpha_c$, $\alpha_{tv}$, and $\alpha_{l_2}$ for MNIST as the private dataset, $\epsilon = 10$, and TinyImageNet and Gaussian Noise as the public dataset initialization.

| Loss | Dataset | Hyperparameter Value and Accuracy | | |
|---|---|---|---|---|
| | | $\alpha_f = 0.1$ | $\alpha_f = 1.0$ | $\alpha_f = 10.0$ |
| $R_{feature}$ | TinyImageNet | $93.99 \pm 0.30$ | $94.56 \pm 0.15$ | $94.03 \pm 0.64$ |
| | Gaussian Noise | $89.55 \pm 0.92$ | $90.30 \pm 1.43$ | $88.53 \pm 0.33$ |
| | | $\alpha_c = 0.01$ | $\alpha_c = 1.0$ | $\alpha_c = 10.0$ |
| $R_{classif}$ | TinyImageNet | $94.42 \pm 0.31$ | $94.03 \pm 0.64$ | $94.49 \pm 0.04$ |
| | Gaussian Noise | $90.99 \pm 0.86$ | $88.53 \pm 0.33$ | $90.81 \pm 0.83$ |
| | | $\alpha_{tv} = 2e\text{-}5$ | $\alpha_{tv} = 0.01$ | $\alpha_{tv} = 10.0$ |
| $R_{tv}$ | TinyImageNet | $94.03 \pm 0.64$ | $94.27 \pm 0.42$ | $94.32 \pm 0.12$ |
| | Gaussian Noise | $88.53 \pm 0.33$ | $89.63 \pm 1.90$ | $89.81 \pm 2.16$ |
| | | $\alpha_{l2} = 3e\text{-}8$ | $\alpha_{l2} = 0.0001$ | $\alpha_{l2} = 0.01$ |
| $R_{l2}$ | TinyImageNet | $94.03 \pm 0.64$ | $94.45 \pm 0.10$ | $94.53 \pm 0.16$ |
| | Gaussian Noise | $88.53 \pm 0.33$ | $90.93 \pm 1.33$ | $89.87 \pm 1.01$ |

Table 7: Values of scaling coefficients for the loss terms $\mathcal{R}_{feature}$, $\mathcal{R}_{classif}$, $\mathcal{R}_{tv}$, and $\mathcal{R}_{l_2}$ in Equation 5 in the main paper: $\alpha_f$, $\alpha_c$, $\alpha_{tv}$, and $\alpha_{l_2}$.

| $\alpha_f$ | $\alpha_c$ | $\alpha_{tv}$ | $\alpha_{l_2}$ |
|---|---|---|---|
| 10 | 1 | 2.5e-5 | 3e-8 |

### A.1.4 Student Model Training on DP-ImgSyn Synthetic Images Hyperparameters

We use Stochastic Gradient Descent (SGD) optimizer (eon Bottou, 1998) with a learning rate $\eta = 0.1$, momentum 0.9, and weight decay 1e-4 for training a student model on the synthetic images. We use the multistep learning rate scheduler with $\gamma = 0.1$ and milestones at 120, 150, and 180 epochs. We train the models for 200 epochs with 256 as batch size.

The optimization loss function is the KL-divergence between the soft labels of the teacher model and the soft labels of the student model, defined as:

$$\min_{\theta} \sum_{x \in \mathcal{X}^s} KL(p_{\mathcal{M}}(\hat{x}), p_{\mathcal{S}}(\hat{x})/T) \tag{7}$$

Where $KL$ refers to the Kullback-Leibler divergence, $p_{\mathcal{M}}(\hat{x})$ and $p_{\mathcal{S}}(\hat{x})$ are the output distributions (soft labels) of the teacher and the student model respectively when the synthetic image $\hat{x}$ is given as input. The temperature value used in our simulations is $T = 100$ for MNIST and FashionMNIST and $T = 10$ for CelebA-Hair and CelebA-Gender.

### A.2 Loss Term Ablation Study

In this section, we evaluate the effect of each loss term on the accuracy of the model: $\mathcal{R}_{feature}$, $\mathcal{R}_{classif}$, $\mathcal{R}_{tv}$, and $\mathcal{R}_{l_2}$. The loss term ablation study is summarized in Table 8. We present the results for MNIST as private set with epsilon=10 when using TinyImageNet, Places365, and FashionMNIST as public datasets using three

Table 8: Ablation study on the loss terms for MNIST as private dataset, $\epsilon = 10$, and TinyImageNet, Places365, FashionMNIST and Gaussian Noise as initialization.

| $\mathcal{R}_{feature}$ | $\mathcal{R}_{classif}$ | $\mathcal{R}_{tv}$ | $\mathcal{R}_{l_2}$ | TinyImageNet | Places365 | FashionMNIST | Gaussian Noise |
|:---:|:---:|:---:|:---:|:---:|:---:|:---:|:---:|
| ✓ | ✗ | ✗ | ✗ | $94.30 \pm 0.25$ | $93.74 \pm 0.13$ | $94.02 \pm 0.09$ | $90.07 \pm 1.59$ |
| ✓ | ✓ | ✗ | ✗ | $94.31 \pm 0.09$ | $93.73 \pm 0.24$ | $94.19 \pm 0.25$ | $89.42 \pm 2.36$ |
| ✓ | ✓ | ✓ | ✗ | $94.34 \pm 0.31$ | $93.74 \pm 0.17$ | $93.94 \pm 0.28$ | $89.82 \pm 1.00$ |
| ✓ | ✓ | ✓ | ✓ | $94.03 \pm 0.64$ | $93.74 \pm 0.18$ | $93.90 \pm 0.30$ | $88.53 \pm 0.33$ |
| ✗ | ✓ | ✓ | ✓ | $93.03 \pm 0.63$ | $93.32 \pm 0.30$ | $91.70 \pm 2.24$ | $63.48 \pm 13.62$ |

different seeds. To get further insight, we also present results when using Gaussian Noise with mean 0 and standard deviation 1 as initialization for the synthetic images. We make the following observations: 1) from the Gaussian Noise results we see that the feature loss that uses the batch normalization statistics significantly affects the accuracy ($\approx 26\%$ accuracy drop when feature loss was excluded in the synthesis loss). 2) when all the loss terms were used this led to a lower standard deviation between the seeds and thus more stability between the runs. The significant effect of the feature loss can be attributed to the value of the scaling coefficient associated with it.

### A.3 Visualizations of ImgSyn Using a Real Example

In this section, we present a real example of Figure 1. In Figure 5 we show how ImgSyn is able to better sample the latent space and improve the decision boundary of the student model. For this experiment, we generated 2D data by sampling from a Uniform distribution in the range $[0, 1]$. The labels were set to 0 or 1 based on the cubic $y = -1.4x^3 + 0.9x1^2 + 0.34$. That is the cubic formed the decision boundary, if $y > 0$ label $= 1$ and vice versa. The cubic was chosen so as to have a non-linear decision boundary. Since the data is 2D we can visualize the decision boundary in the input space. The top left in Figure 5 shows the training data, the two classes shown in green (class 1) and orange (class 2). The teacher model was trained on this classification problem. Please note no DP was used during training since we want to demonstrate ImgSyn. The network confidence is visualized as a heatmap. Blue is high confidence that the sample is in class 1 and yellow is high confidence that the sample is in class 2. The top right shows the same visualization for the student model when applying vanilla knowledge distillation (KD) between the teacher and the student model. It shows the decision boundary at the end of training and the data (illustrated in red) that was used for performing knowledge distillation. For this example, we used two Gaussian clusters centered at $(0.25, 0.25), (0.75, 0.75)$ with a deviation of 0.25 as the dataset to perform knowledge distillation from the teacher to the student. Clearly vanilla KD is not able to transfer the decision boundary well from the teacher to the student. The bottom left shows the data after applying our ImgSyn on the same Gaussian dataset to align it with the source distribution, and the teacher decision boundary when passing the ImgSyn generated data. We see how the points move closer into the range of $[0, 1]$ to match the source distribution and on the bottom right we show the result of training a student model on the ImgSyn data. Clearly the decision boundary of the student model trained on ImgSyn data has a better match to the teacher model than the student trained with vanilla KD on Gaussian data.

### A.4 Interference of the features between Public and Private Images

In this section, we quantify the interference of the public image features on the student model test accuracy. To evaluate this, we perform the following experiment: we initialize the synthetic images with Gaussian noise and then perform DP-ImgSyn, using MNIST as private dataset and epsilon equal to 10. This will set the lower bound on the interference that the public dataset has on the optimization since the starting images are random. We evaluate the accuracy of the student model trained on the synthetic images initialized with Gaussian noise (Table 9). We observe that the accuracy drops about 3.4% for the DP-ImgSyn with Gaussian noise initialization compared to best-performing public images. Even though there is interference from the public set, this effect is minimal compared to the effect of DP-ImgSyn. Without DP-ImgSyn, the student

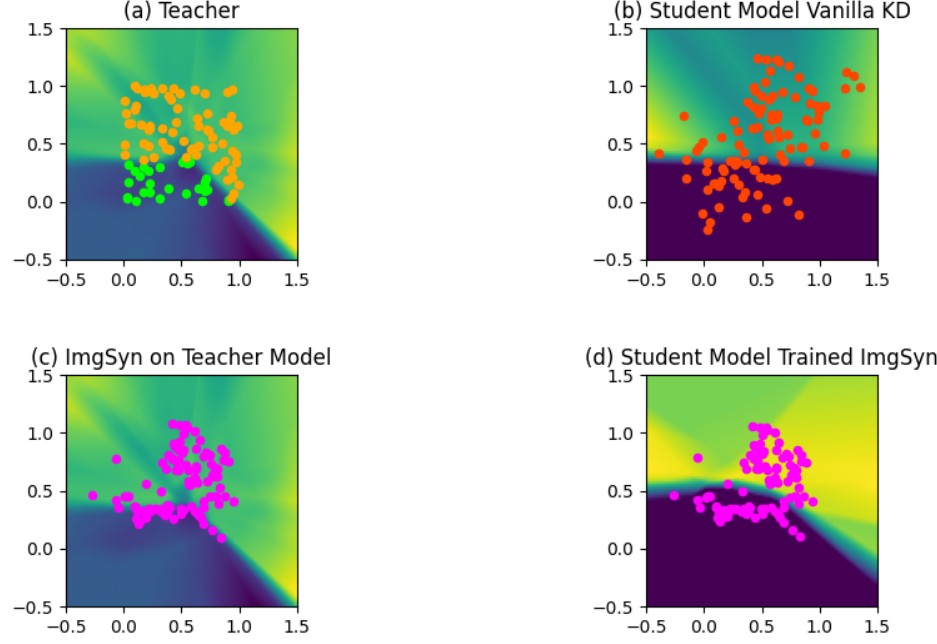

Figure 5: Example visualizing deep neural networks decision boundary when using ImgSyn and not using ImgSyn, a data-driven version of Figure 1: (a) visualization of the teacher decision boundary, and the training data belonging to two classes. The two classes are shown in green (class 1) and orange (class 2). The teacher network confidence is visualized as a heatmap. Blue illustrates high confidence that the sample is in class 1 and yellow is high confidence that the sample is in class 2, (b) visualization of the decision boundary for the student model when applying vanilla KD between the teacher and the student model, and the Gaussian data used for the vanilla KD (visualized in red) this is analogous to public data, (c) visualization of the teacher decision boundary and the ImgSyn generated data (magenta). The ImgSyn data are generated after applying our ImgSyn on the Gaussian data (illustrated in red color in image (b)) to align it with the source distribution, (d) visualization of the decision boundary of the student model trained on ImgSyn data, and the ImgSyn data (magenta) used for training the student model. The decision boundary of the student model trained on ImgSyn data has a better match to the teacher model than the student trained with on Gaussian data.

Table 9: The interference of the features of the public images on the private images. The private dataset is MNIST, $\epsilon = 10$. The results for the datasets marked with * are from the main paper (Table 2), and are included in this table as a reference for comparison with the Gaussian Noise.

| Private Dataset | $\epsilon$ | Initialization | DP-ImgSyn(0) | DP-ImgSyn(k) |
|---|---|---|---|---|
| MNIST | $\epsilon = 10$ | Gaussian Noise | $30.12 \pm 5.08$ | $\mathbf{90.55 \pm 1.67}$ |
| | | TinyImageNet* | $92.97 \pm 0.65$ | $\mathbf{94.03 \pm 0.64}$ |
| | | Places365* | $92.63 \pm 0.23$ | $\mathbf{93.74 \pm 0.18}$ |
| | | FashionMNIST* | $93.61 \pm 0.37$ | $\mathbf{93.90 \pm 0.30}$ |

model trained using Gaussian noise achieves 30.12% accuracy on MNIST while with DP-ImgSyn it achieves 90.55% (Table 9).

Table 10: Comparison Table with state-of-the-art techniques for $\epsilon = 0.2$ for MNIST and FashionMNIST and TinyImageNet as the public dataset initialization. Results for DP-ImgSyn are mean over three different seeds. DP-GAN refers to Xie et al. (2018), DP-MERF refers to Harder et al. (2021), P3GM refers to Takagi et al. (2021), DataLens refers to Wang et al. (2021) and G-PATE refers to Long et al. (2021).

|  | DP-GAN | DP-MERF | P3GM | DataLens | G-PATE | DP-ImgSyn (Our) |
|---|---|---|---|---|---|---|
| MNIST | 11.04% | 62.61% | 8.20% | 23.44% | 22.30% | 77.37% |
| FashionMNIST | 10.21% | 52.61% | 12.80% | 22.26% | 18.74% | 70.63% |

Table 11: Gaussian Noise with low-pass filtering as initialization with MNIST as the private dataset and $\epsilon = 10$. The low pass filtering is implemented as a Gaussian blur Haddad et al. (1991) for various kernel sizes.

| Without Low Pass Filtering | Low-pass Filtering | | | |
|---|---|---|---|---|
|  | Kernel Size = 3 | Kernel Size = 7 | Kernel Size = 11 | Kernel Size = 15 |
| $90.55 \pm 1.67$ | $91.62 \pm 1.35$ | $90.82 \pm 0.96$ | $91.10 \pm 0.41$ | $91.18 \pm 1.23$ |

### A.5 Comparison With SOTA under Strong Privacy

In this section, we present experiments with MNIST and FashionMNIST as private datasets with an epsilon value of 0.2 and TinyImageNet as the public dataset initialization. Table 10 summarizes the comparison results. Our method demonstrates better accuracy than the best-performing SOTA (DP-MERF) by 14.76% on MNIST and 18.02% on FashionMNIST.

### A.6 Gaussian Noise Initialization with Low-pass Filtering

In this section, we present the results when Gaussian noise is used as initialization with MNIST as the private dataset and $\epsilon = 10$, and then we apply low pass filtering implemented as a Gaussian blur Haddad et al. (1991). The results for various kernel sizes are summarized in Table 11. Notably, low pass filtering improves the accuracy of the Gaussian noise initialization by up to 1.07% (results for kernel size 3). However, it is still lower ($\approx 2.4\%$) than the best-performing public dataset initialization. Therefore, random noise can as initialization when public images are unavailable, yielding satisfactory performance albeit with some accuracy degradation.

### A.7 More Visualizations

Figure 6 provides illustrations for MNIST, FashionMNIST, and CelebA Gender as private images, the DP-ImgSyn generated images and the corresponding public images and Gaussian Noise that were used as initialization for DP-ImgSyn.

### A.8 FID Score Evaluation

We evaluate our proposal using the Frechet inception distance (FID) score. The FID score is calculated based on the feature (latent space) representations extracted from a pretrained deep neural network (Inception-v3 model), which has been trained on a large and diverse image dataset (ImageNet). From Table 12, we observe that our proposal achieves a low FID score (lower is better) and outperforms SOTA works. Following SOTA works Wang et al. (2021), the FID score is not recommended as a metric for grayscale images (MNIST,

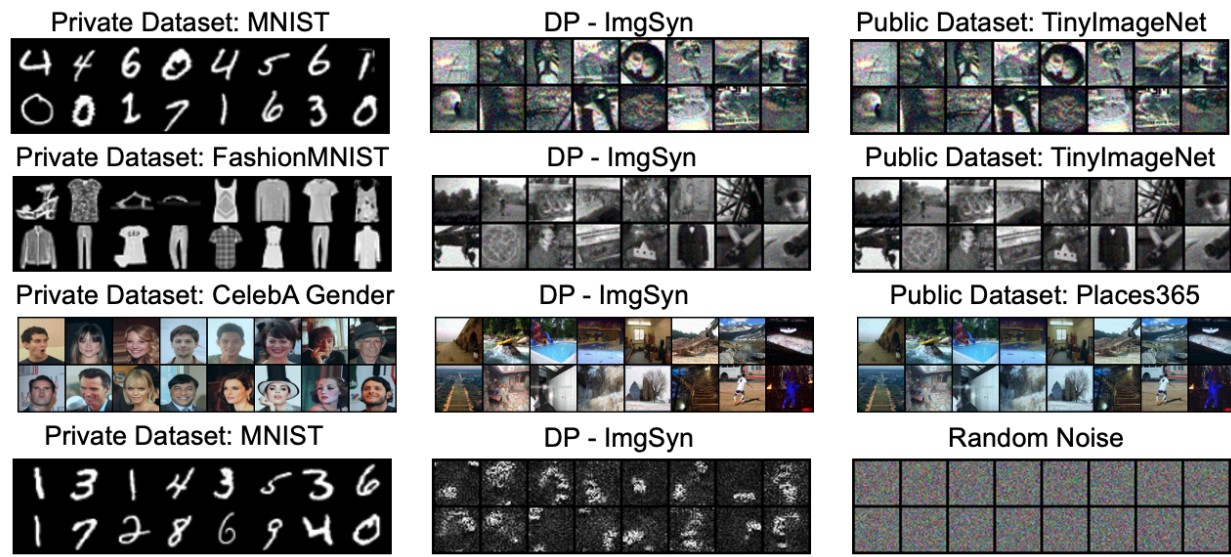

Figure 6: We visualize the private, synthetic, and public images (from left to right) for: (first row) MNIST as private dataset with TinyImageNet as public dataset (on TinyImageNet we apply grayscale image transformation because MNIST images are grayscale), (second row) FashionMNIST as private dataset with TinyImageNet as public dataset (on TinyImageNet we apply grayscale image transformation because FashionMNIST images are grayscale), (third row) CelebA Gender as private dataset with Places365 as public dataset, (forth row) MNIST as private dataset with Gaussian Noise with zero mean and one standard deviation as initialization.

Table 12: SOTA comparison using FID score for CelebA dataset with Places365 (A) and LSUN (B) as public datasets for DP-ImgSyn.

| Methods | DP-GAN | DP-MERF | P3GM | DataLens | G-PATE | DP-ImgSyn (ours) |
|---------|--------|---------|------|----------|--------|------------------|
| $\epsilon$ | $10^4$ | $10^4$ | $10^4$ | 10 | 10 | 10 |
| FID $\downarrow$ | 403.94 | 327.24 | 435.60 | 320.84 | 305.92 | **188.62 (A), 194.88 (B)** |

FashionMNIST) because it involves a pretrained network on RGB images. This affects the evaluation of the metric and the resulting evaluation is not meaningful, as explained in Wang et al. (2021).

## A.9 Dataset Statistics

Table 13 summarizes the statistics for the vision datasets used for the experimental evaluation in Section 4 in the main paper: MNIST, FashionMNIST, CIFAR-10, Imagenette, CelebA-Hair, CelebA-Gender, TinyImageNet, Places365, LSUN, and Textures. We report the train and test size, the resolution of the images, and the number of classes in each dataset.

Table 13: Dataset statistics, training, and test set sizes for the datasets used.

| Dataset | Train Set Size | Test Set Size | Resolution | Number of Classes |
|---|---|---|---|---|
| MNIST | 60,000 | 10,000 | 28x28 | 10 |
| FashionMNIST | 60,000 | 10,000 | 28x28 | 10 |
| CIFAR-10 | 50,000 | 10,000 | 32x32 | 10 |
| Imagenette | 10,000 | 5,000 | 224x224 | 10 |
| CelebA-Hair | 162,770 | 19,962 | 64x64 | 3 |
| CelebA-Gender | 162,770 | 19,962 | 64x64 | 2 |
| TinyImageNet | 100,000 | 10,000 | 32x32 | 200 |
| Places365 | 1,803,460 | 10,000 | 32x32 | 365 |
| LSUN | 9,895,373 | 303,304 | 64x64 | 10 |
| Textures | 5,640 | 1,880 | 224x224 | 47 |

