# OpenReview forum: "DP-ImgSyn: Dataset Alignment for Obfuscated, Differentially Private Image Synthesis"
_TMLR — Rejected by TMLR_

### Review · Reviewer_NUvT · 2023-11-24

**Summary Of Contributions:**

This paper proposes a new differentially private image synthesis method by aligning public images to private images in the latent space of a private teacher model.

**Audience:**

Yes

**Claims And Evidence:**

No

**Requested Changes:**

Please refer to the Weaknesses

**Strengths And Weaknesses:**

Strengths:
1. The proposed idea is interesting in general, by aligning public images to the private domain.
2. The paper is well written.

Weaknesses:
1. The major problem of this paper is the motivation. As in Figure 5, the generated images are very different from the private dataset, which means they have a low "quality". In conventional DP image generation, we usually want to have better both "utility" and "quality". The proposed method only focuses on the "utility". However, if we only care about "utility", why not just release the private teacher model/classifier (guaranteed by DP)? Why do we still need to release the generated dataset in the setting?
2. What is the impact of the domain gap between the public dataset and the private dataset? If the private dataset is very different from public data, which can be a common case in the real world, will this method still be useful?
3. In algorithm 1, how are target labels selected? Is there any relationship between the target label and x_P?
4. Figure 3 is in low resolution.

---

> ### Author Response · Authors · 2023-12-01
>
> We thank the reviewer for the feedback and address the identified weaknesses sequentially.
>
> 1.	We would like to clarify that the aim of the paper is to generate synthetic images that are visually dissimilar to the private images (Figures 2 and 6 in the revised version) and achieve utility (classification accuracy), as the results in Tables 2 and 3 indicate. We have updated the introduction to detail the task under consideration. Note, these synthetic images can be publicly released without revealing the private images. Our approach is different from the conventional GAN-based differential privacy (DP) image generation which result in synthetic images that are visually like the private images. The deliberate visually dissimilar image generation aligns with our specific use case, which involves scenarios where the content of images may be disturbing or sensitive, such as in content moderation tasks. In these contexts, the desired outcome is not only to preserve utility (classification accuracy) but also to achieve visual dissimilarity between the private and the synthetic images. Our method is positioned as a distinct strategy within the broader spectrum of privacy-preserving techniques. We believe that our approach contributes a valuable perspective to the field by addressing the nuanced challenges associated with visual content in private datasets. Moreover, there are two main approaches for privacy preserving machine learning: 1) releasing data, and 2) releasing models. We would like to clarify that our proposal falls in the category of releasing data. Releasing the data rather than models has more scope for improvement: by combining data from different sources, better models can be built. The user is not bounded by the architecture of the released classifier but can use any desired model architecture. Also, it reduces or eliminates catastrophic forgetting and issues with continual learning when more data is available to train the classifier further [1,2].
>
> [1] Kemker, Ronald, Marc McClure, Angelina Abitino, Tyler Hayes, and Christopher Kanan. "Measuring catastrophic forgetting in neural networks." In Proceedings of the AAAI conference on artificial intelligence, vol. 32, no. 1. 2018.
>
> [2] De Lange, Matthias, Rahaf Aljundi, Marc Masana, Sarah Parisot, Xu Jia, Aleš Leonardis, Gregory Slabaugh, and Tinne Tuytelaars. "A continual learning survey: Defying forgetting in classification tasks." IEEE transactions on pattern analysis and machine intelligence 44, no. 7 (2021): 3366-3385.
>
> 2.	The impact of domain gap (accuracy) between the public and the private images is minimal. This is seen from Table 2, which used various public datasets and they resulted in comparable accuracy. Furthermore, we provide a study about interference of the features between public and private images in the Appendix of the paper (Section A.4). As it is mentioned in the paper, for this analysis we initialized the synthetic images with Gaussian Noise, which has no prior visual information (MNIST was used as private dataset with epsilon=10). The results of this are presented in Appendix Table 9. We observe that the accuracy drops about 3.4% for the DP-ImgSyn with Gaussian noise initialization compared to best-performing public images. Even though the choice of the public set has some effect, this effect is minimal compared to the effect of DP-ImgSyn. Without DP-ImgSyn, the student model trained using Gaussian noise achieves 30.12% accuracy on MNIST while with DP-ImgSyn it achieves 90.55%. Moreover, we would like to clarify that aim of our paper is to achieve visual dissimilarity between the public and the private images, while having high classification accuracy. Figure 2 illustrates the private dataset (human faces) and the public images (places), which are visually dissimilar. Table 2 reports the classification results which indicate the effectiveness (classification accuracy) of our method.
>
> 3.	The target labels are assigned such that we have the same number of images for each class in the synthetic dataset as in the private dataset, as mentioned in Section 3.2 of the paper. x_P (public image) is any image selected from the public dataset, and no specific constraints are imposed between the target label and the associated public images. Specifically, if n samples are being synthesized and the private set is a K class classification problem, we generate n/K samples for each class. The targets are independent of the private data, thus leaks no privacy. For further transparency, we have provided the exact PyTorch implementation in the appendix in Section A.1.3 (the code is also provided in the supplementary material).
>
> 4.	We will improve the resolution (we have made it a vector rather than a raster image) of Figure 3 to enhance visual comprehension for the readers.

---

### Review · Reviewer_wQQR · 2023-12-04

**Summary Of Contributions:**

The paper presents a Differentially Private Image Synthesis approach to sanitize and release sensitive image data. Downstream applications can train a classifier on the synthesized data, which can operate on sensitive data. The key idea is to take a public dataset, and perturb it such that the decision boundary on the perturbed data is the same as that of the sensitive dataset.

Results on multiple datasets and comparisons against existing generative approaches demonstrate the effectiveness of the proposed approach.

**Audience:**

Yes

**Broader Impact Concerns:**

The reviewer does not see any ethical concerns for this work. So a broader impact statement is not necessary.

**Claims And Evidence:**

Yes

**Requested Changes:**

I first have one question on this line of research.
- The implicit goal of the paper was to release realistic images. Hence a public image dataset was chosen and perturbed to match the statistics of the private data. Since the images are being released for machine consumption, why can't this be done with random images? Why should we do this with a public dataset? If I understood them correctly, the experiments in the appendix show that Gaussian noise-initialized data performs comparably. Perhaps with a bit of low-pass filtering, the performance would go up even further.
- Why is a TV loss required? Why does the resulting image have to be smooth, or even look like a real image?

The following is a question more specific to the paper:
- On page 2, there is a sentence, "This is only possible in a non-generative framework". This is a strong claim, can you elaborate?
- Perhaps this information was provided and I missed it. During evaluation is the student model evaluated on the synthesized test data or the private data? I would imagine that the goal is to train the student model on the synthetic data, but eventually evaluate how useful it is for classifying real private images. Please clarify the experimental setup.
- Apart from training models for classifying the images, what other use cases are there for the released synthetic dataset? Can it be used for other tasks, say semantic segmentation, object detection, etc.? How does the classification objective in the image synthesis process limit the possible downstream applications of the synthesized data?
- Conceptually why is the $\ell_2$ loss required?
- The variance in Table 7 is larger than the difference in means. Looks like feature loss is sufficient. Could you comment on this?
- Lastly, can you initialize from generated data and align them just in the way the public dataset was aligned? I imagine this will improve the  results of the generated data too.

**Strengths And Weaknesses:**

Strengths:
- The idea of the paper is interesting. Adopting a public dataset and perturbing it to match the statistics of the private data and then releasing the perturbed data.
- The classification results on the perturbed data demonstrate the effectiveness of the proposed method. In almost all cases, the proposed approach leads to better classification results.


Weaknesses: The paper is pretty solid, did not find too many, except for a few minor concerns.
- There are four hyper-parameters to control the total loss. There does not seem to be much discussion of the stability of the model to these hyper-parameters.
- The $\epsilon$ value chosen is on the larger side (either 1 or 10). Don't think these values provide practical privacy. There is no discussion on smaller values of $\epsilon$.

---

> ### Author Response · Authors · 2023-12-12
>
> We thank the reviewer for the feedback. In response, we address the identified weaknesses:
>
> •	We added a table (Table 6 in the revised paper) for the four hyper-parameters that control the total loss on MNIST as private dataset for epsilon 10, and TinyImageNet as the public dataset initialization. Specifically, we search the values for each loss scaling factor while keeping the remaining scaling factors constant. From the results in Table 6, we observe that our method is robust to the scaling factor hyperparameter selections. Furthermore, we conducted an experiment using the same setup (MNIST as private dataset with epsilon set to 10), but with Gaussian noise as initialization. The findings consistently affirm our earlier observation, highlighting the robustness of our method to different scaling factor hyperparameter selections. This analysis has been incorporated into the revised paper.
>
> •	We conducted experiments with MNIST and FashionMNIST as private datasets with an epsilon value of 0.2 and TinyImageNet as the public dataset initialization. The comparison table is provided in the revised version (please see Table 10). Our method demonstrates better accuracy than the best-performing SOTA (DP-MERF) by 14.76% on MNIST and 18.02% on FashionMNIST. These results have been incorporated into the revised paper.
>
> We answer the questions related to the line of research:
>
> •	 In our experiments we use public images as initialization, showcasing the effectiveness of our method. When no public images are available, then we can use as initialization Gaussian noise (refer to Table 9 in the Appendix). The outcome of these experiments was that “the accuracy drops about 3.4% for the DP-ImgSyn with Gaussian noise initialization compared to best-performing DP-ImgSyn with public images”, as mentioned in the paper. Additionally, we performed the same experiment, i.e. the images are initialized with Gaussian noise, and then we apply a low pass filtering implemented as a Gaussian blur [1]. The results for various kernel sizes are summarized in Table 11 (revised version). Notably, low pass filtering improves the accuracy of the Gaussian noise initialization up to \~1% (results for kernel size 3). However, it is still lower (\~2.4%) than the best-performing public dataset initialization. Therefore, Gaussian noise can serve as an initialization method when public images are unavailable, yielding satisfactory performance albeit with some accuracy degradation. Furthermore, the additional application of low-pass filtering can help mitigate the accuracy drop to a certain extent. This analysis has been incorporated in the revised paper.
>
>  [1] Haddad, Richard A., and Ali N. Akansu. "A class of fast Gaussian binomial filters for speech and image processing." IEEE Transactions on Signal Processing 39, no. 3 (1991): 723-727.
>
> • 	The resulting synthetic image is not necessarily required to be smooth or resemble a real image. However, we observed experimentally that when including the TV loss, it resulted in lower variance in model performance (in Table 8 – see rows 2 and 3 where the standard deviation with Gaussian Noise initialization is 1.0 with using the TV loss and 2.36 without using the TV loss).
>
>  Next, we address the questions that are more specific to the paper:
>
> •	We were trying to convey that performing KD and transferring decision boundaries from teacher to student is non-trivial for generative methods. However, we agree with the reviewer that this is a strong claim, thus we have removed it in the revised paper.
>
> •	The student model is trained on the synthetic images, and it is evaluated on the private images, as indicated in the caption of Table 2.
>
> •	In our work, we focus on image classification tasks. This is similar to previous research in this area whose comparisons are provided in the paper. It might be possible to apply this in other tasks in a supervised setting. However, exploring the technique in an unsupervised setting could be an intriguing research direction.
>
> •	The l2-norm is employed to encourage the image range to remain within a target interval rather than diverging. This clarification has been incorporated into the revised paper.
>
> •	When initializing with Gaussian noise, incorporating all the losses results in the lowest standard deviation and, consequently, a more stable performance (see Table 8 in the revised version). If our primary concern is the model's performance, we may choose to exclude the remaining losses and solely utilize the feature loss.
>
> •	Using GAN-generated images as initialization instead of public images for the DP-ImgSyn, could potentially improve the results. However, using GAN-generated images, which closely resemble the private images, would consequently make the synthetic images produced by DP-ImgSyn visually similar to the private images. The motivation of our work is to release images that are visually dissimilar to the private images.

---

### Review · Reviewer_NoVW · 2023-12-06

**Summary Of Contributions:**

The authors have developed a differentially private method for synthesizing image datasets. Their method aligns a public dataset to behave similarly to a private dataset. To achieve this, they construct a differentially private classifier for the private dataset, which they refer to as the 'teacher model,' and develop a loss function based on the trained teacher model. Experimental results demonstrate the superior accuracy of the proposed method compared to existing massive methods.

**Audience:**

Yes

**Broader Impact Concerns:**

I have no concerns regarding border impact.

**Claims And Evidence:**

No

**Requested Changes:**

Weaknesses 1-5 are major, so I would like the authors to address them. Weakness 6 is minor.

**Strengths And Weaknesses:**

### Strengths

1. The paper's strength lies in its comprehensive comparison with various differentially private image synthesizing methods, demonstrating the superior accuracy of the proposed method.

### Weakness

1. A notable weakness of the paper is its lack of a clear formulation of the task. It appears to discuss the semi-private classification problem, a task introduced in the following paper, within the methodology and experimental results sections:
- Alon et al. Limits of Private Learning with Access to Public Data. NeurIPS19.

In the initial sections, including the title, the focus is on a differentially private image synthesis task, which aims to generate images that resemble private ones in a privacy-preserving manner. This leads to confusion due to the lack of a formal definition and clear task delineation.

2. If the targeted task is semi-private classification, the paper lacks comparisons with existing works in the field. Many relevant papers in this area are not cited, including, but not limited to, the following papers and the references contained therein:
- Bie et al. Private Estimation with Public Data. NeurIPS22.
- Ganesh et al. Why Is Public Pretraining Necessary for Private Model Training? ICML23.

(While these papers focus on theoretical analyses, their references include papers that empirically evaluate semi-private classification.)

3. If the targeting task is the private image synthesizing, the paper lacks critical evaluations, such as those on the quality of the synthesized images and performance in other downstream tasks. It is crucial to assess these areas, as the image synthesis method may be incorporated into various tasks.

4. The privacy guarantee of the proposed algorithm raises concerns. The classification loss mentioned in Eq. (2) includes the target label y, which seems to be derived from the private dataset. This inclusion could lead to a potential breach of privacy.

5. The explanation of the algorithm, particularly in relation to Fig. 1, is not convincingly demonstrated. The process of minimizing the loss defined in Eqs. 1-5 does not appear to directly cause the point movements shown in Fig. 1. Moreover, the experiments do not validate or illustrate this behavior effectively. This gap in the explanation and empirical evidence regarding Fig. 1 should be addressed to strengthen the argument.

6. The scaling temperature parameter in Eq. 6 may be meaningless, as multiplying a positive constant by the objective function does not change the optimal solution.

---

> ### Author Response · Authors · 2023-12-12
>
> 1.	We thank the reviewer for identifying the need for delineating the task under consideration. To clarify this to the readers we have updated the introduction with a specific section to emphasize the task considered.  Specifically, we have added, “we consider the following task of generating a synthetic dataset with the following three requirements (1) The synthetic dataset must be (ε, δ) differentially private. (2) The synthetic dataset must be visually dissimilar to the public dataset i.e. generating visually non-sensitive when the private dataset is such. (3) The synthetic dataset has a similar utility as the private dataset in the downstream task of classification”. As mentioned in the introduction a strong use case involves scenarios where the content of images may be disturbing or sensitive, such as in content moderation tasks. In such cases generating more of such data raises some ethical questions. Thus, in our method we propose generating images that are visually dissimilar to the source distribution while maintaining the utility of the data.
>
> 2.	We thank the reviewer for drawing parallels with semi-private learning, however the task described in this paper is quite different from semi-private learning. While many semi-private learning techniques exist, the spirit of semi-private learning is to leverage public data to improve the privacy bound when learning the private data. Quantitatively we interpret this as updating the private model’s parameters with public data to improve privacy guarantees to have a better privacy-utility trade off. As mentioned in our paper, public data is NOT used during TRAINING to improve privacy utility trade off.  Thus, papers related to semi-private learning are not relevant in our case. Further we wish to re-emphasize that we focus on differentially private image synthesis with visual dissimilarity.
>
>
> 3.	Regarding downstream tasks, our method targets image classification only. We provide very thorough evaluations of the downstream classification task. This is similar to previous research in this area whose comparisons are provided in the paper. We refer the reviewer to the results of using the dataset on downstream classification which are provided in Tables 2, 3 and section 4.5. Further regarding FID / image quality metrics our explicit goal is to be visually different from the source distribution, but we do provide FID scores in Table 12 (in the revised Manuscript, Table 9 in the original submission) for image synthesis metrics.
>
> 4.	We realize that the writing here is confusing and have clarified how the target label is generated. If n samples are being synthesized and the private set is a K class classification problem, we generate n/K samples for each class. The labels are uniformly distributed over the n samples and the private dataset’s label are NOT used. Hence, we believe this does not leak any private information. For further transparency we have provided the exact PyTorch implementation in the appendix (the code was (and is) also provided in the original supplementary material).
>
>
> 5.	We agree that visualization is relevant to validating Figure 1. We would like to emphasize that the results on all the datasets are in line with the explanations in Figure 1. To address the visualization gap, we have added a visualization section (for illustration purposes) in the appendix (See Figure 5 in the revised version) that trains a deep neural network with and without DP-ImgSyn and shows the movement of the data points under DP-ImgSyn and improved decision boundary. We have also included a video of the entire training process visualizing the decision boundaries and image synthesis (where we clearly see data movement) in the supplementary material.
>
> 6.	We thank the reviewer for identifying this typo. The T should be within the parenthesis. It has been updated in the revised manuscript.

---

### Decision · Action_Editor_m7Tw · 2024-01-12

**Recommendation:** Reject

**Comment:**

This paper has developed a differentially private method for synthesizing image datasets. The proposed method aligns a public dataset to behave similarly to a private dataset. To achieve this, the authors construct a differentially private classifier for the private dataset, which they refer to as the teacher model, and develop a loss function based on the trained teacher model. Experimental results demonstrate the superior accuracy of the proposed method compared to existing massive methods.

The idea of this paper is nontrivial enough to intrigue other TMLR audiences. However, the current version still has some problems. For example, 1) The major issue of this paper is the absence of clear evidence supporting their claim and the unrealistic setup of the problem. In the revised manuscript, the authors describe their task as generating a private synthetic dataset, which must be dissimilar to the public dataset. However, they fail to provide any evidence or experimental results confirming that the dataset produced by their algorithm is dissimilar to the public dataset. Indeed, the dataset appears similar to the public dataset in the visualized images in Figure 6. 2) Furthermore, the authors contend that the public dataset is not available during the training phase. This assumption is unrealistic since a public dataset is typically expected to be accessible on demand. 3) The problem is not well-defined, and as such the purpose of the proposed method is unclear.

Therefore, we cannot accept this work this time, but the authors are encouraged to resubmit after a major and significant revision. We will consider to recommend its acceptance if the authors had addressed these issues properly.

**Audience:**

Yes

**Claims And Evidence:**

The current version still has some problems. For example, 1) The major issue of this paper is the absence of clear evidence supporting their claim and the unrealistic setup of the problem. In the revised manuscript, the authors describe their task as generating a private synthetic dataset, which must be dissimilar to the public dataset. However, they fail to provide any evidence or experimental results confirming that the dataset produced by their algorithm is dissimilar to the public dataset. Indeed, the dataset appears similar to the public dataset in the visualized images in Figure 6. 2) Furthermore, the authors contend that the public dataset is not available during the training phase. This assumption is unrealistic since a public dataset is typically expected to be accessible on demand. 3) The problem is not well-defined, and as such the purpose of the proposed method is unclear.

**Resubmission Of Major Revision:**

The authors may consider submitting a major revision at a later time.

---

> ### Author Response · Authors · 2024-02-06
>
> We thank the action editor for the feedback. We would like to clarify that there was a typo in the second page of the paper related to the core of our work (public/private/synthetic datasets and similarity/dissimilarity). We wrote that: “Specifically, we consider the following task of generating a synthetic dataset with the following three requirements (1) The synthetic dataset must be (ϵ, δ) differentially private. (2) The synthetic dataset must be visually dissimilar to the **public** dataset, i.e., generating visually non-sensitive when the private dataset is such. (3) The synthetic dataset has a similar utility as the private dataset in the downstream task of classification.”. However, the correct statement is: “Specifically, we consider the following task of generating a synthetic dataset with the following three requirements (1) The synthetic dataset must be (ϵ, δ) differentially private. (2) The synthetic dataset must be visually dissimilar to the **private** dataset, i.e., generating ...”. As mentioned by the editor in their feedback, we observe in Figure 6 (of the paper), that the public and the synthetic images are visually similar and the private and the synthetic images are visually **dissimilar**. We understand that this typo created confusion and it will be fixed in the revised version. Please see the figure (https://imgur.com/a/94WSA0c) for the overview. Further, we would like to highlight that our method needs the public images during the image synthesis process. After the synthetic images are generated, they are released to train any network. Thus, the public images are not needed for the network training, as the network is trained using synthetic images. To summarize, we study the following problem statement: private datasets cannot be publicly released when they contain visually disturbing and sensitive content (content moderation images, violent images, etc.). Our method synthesizes a dataset (images) from a public dataset that has DP guarantees, results in similar classification accuracy as the private images, and is visually dissimilar to the private dataset.